# Probabilistic Vision-Language Representation for Weakly Supervised Temporal Action Localization

## ABSTRACT

Weakly supervised temporal action localization (WTAL) aims to detect action instances in untrimmed videos with only video-level annotations. As many existing works optimize WTAL models based on action classification labels, they encounter the task discrepancy problem (*i.e.*, localization-by-classification). To tackle this issue, recent studies have attempted to utilize action category names as auxiliary semantic knowledge with vision-language pre-training (VLP). However, there are still areas where existing research falls short. Previous approaches primarily focused on leveraging textual information from language models but overlooked the alignment of dynamic human action and VLP knowledge in joint space. Furthermore, the deterministic representation employed in previous studies struggles to capture fine-grained human motion. To address these problems, we propose a novel framework that aligns human action knowledge and VLP knowledge in the probabilistic embedding space. Moreover, we propose intra- and inter-distribution contrastive learning to enhance the probabilistic embedding space based on statistical similarities. Extensive experiments and ablation studies reveal that our method significantly outperforms all previous state-of-the-art methods. Our code will be available after publication.

## CCS CONCEPTS

• **Computing methodologies** → **Activity recognition and understanding**.

## KEYWORDS

Video Understanding, Human Action Understanding, Vision Language Pre-training

## 1 INTRODUCTION

The development of multimedia services, such as YouTube and Netflix, has led to increasing interest in the field of computer vision for analyzing long-form videos. Temporal Action Localization (TAL) refers to the problem of precisely determining the time intervals in a lengthy, untrimmed video when human activities occur, which is of fundamental significance in video understanding [15, 30, 31].

Fully supervised TAL [3, 4, 26, 27, 49, 58, 59] handles this task based on frame-level, rich annotations. Despite its great success,

**Unpublished working draft. Not for distribution.**

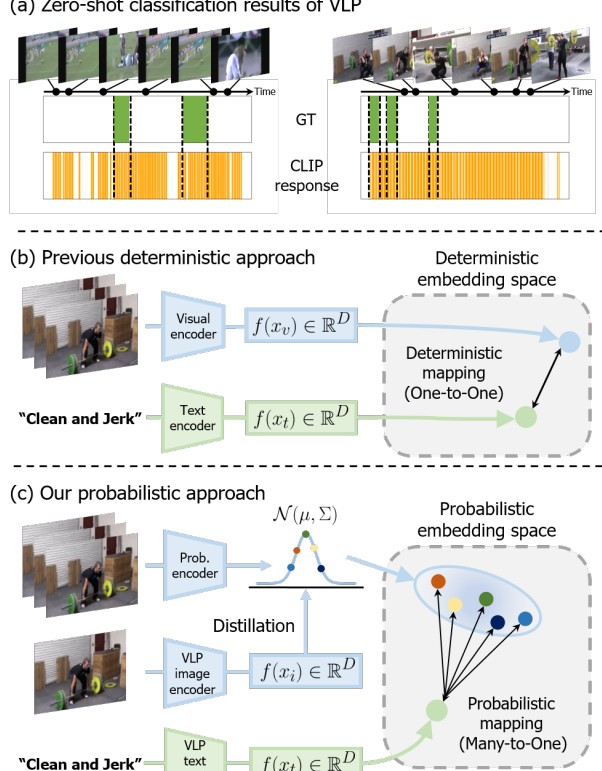

(a) Zero-shot classification results of VLP

(b) Previous deterministic approach

(c) Our probabilistic approach

**Figure 1: (a) CLIP's deterministic pre-training with image-text pairs fails to equip it with the necessary understanding of fine-grained human motion variations. (b) Earlier studies have primarily emphasized the direct mapping between language models and visual input based on deterministic representation. (c) The proposed framework with probabilistic embedding with VLP knowledge alignment.**

training a TAL model in dense frame-level annotation poses challenges in that the cost of annotation is extremely high and its generality is limited. To address these challenges, weakly supervised temporal action localization (WTAL) [9, 11, 18, 21, 24, 33, 45], which only requires video-level categorical labels, has received a lot of attention. In scenarios where only video-level category supervision is accessible, existing WTAL methods solve the localization problem as a classification problem that selects discriminative snippets[1] contributing mainly to video-level classification. However, these classification-based approaches mainly suffer from the task discrepancy problem, which results from a localization-by-classification framework. To deal with this problem, a lot of research has been

---

[1]In our field, a snippet refers to a set of consecutive frames composed of 16 frames.

done on snippet-level pseudo label approaches [12, 25, 42, 65]. Yet, these pseudo labels, constrained by video-level annotations, inherently carry noisy proposals and fall short of achieving the desired accuracy.

Recently, to tackle these problems, some approaches [17, 24] leveraging action category text information have been proposed to guide powerful semantic knowledge without incurring extra annotation costs. These approaches establish additional learning cues by exploiting category text embedding vectors instead of merely utilizing category information as one-hot vectors. While significant improvements were achieved through additional semantic information, some factors were overlooked in earlier research. **First,** Li *et al.* [24] adopted a language model (*e.g.*, GloVe [38]) pre-trained on only text modality, which results in an inadequate initialization state with respect to the alignment with the human action pre-trained visual feature. **Second,** Chen *et al.* [17] proposed an alternate optimization strategy to introduce an effective distillation framework. However, the proposed alternate optimization scheme necessitates manual identification of the optimal settings in accordance with the dataset. **Third,** most importantly, we observe that the utilization of deterministic representation in previous studies for incorporating text information is not suitable for human action understanding.

To confirm this, we conducted an analysis of the zero-shot classification with CLIP [39], a prominent study in the domain of VLP. As shown in Figure 1(a), we compared the similarity response between the text prompt (*i.e.*, "a frame of [CLS]") representation and the corresponding frame visual representation. It reveals a high level of activation even when actual human actions do not occur, as long as there is visual relevance to the action text category. This is because CLIP was pretrained, considering only one-to-one matching between a single image and its caption. The previous research depicted in Figure 1(b) cannot address the aforementioned issue as it solely relies on deterministic representation via one-to-one matching, making it challenging to capture fine-grained human motion. Furthermore, the lack of consideration for direct alignment with pre-trained human action knowledge results in insufficient temporal dynamics modeling.

To overcome this issue, we introduce a novel framework, **PVLR**, **P**robabilistic **V**ision **L**anguage **R**epresentation for Weakly supervised Temporal Action Localization, which integrates VLP knowledge and human action knowledge within the probabilistic embedding space, as shown in Figure 1(c). To begin with, pre-trained human action knowledge, such as Kinetics [2], is utilized to construct a probabilistic embedding space. In this step, probabilistic adapters are introduced to estimate parameters for the snippet-level probability distribution. Subsequently, we transfer the large-scale VLP knowledge to the estimated probability distribution to create a joint probabilistic embedding space. To capture the temporal dynamics of action, we obtain samples from the estimated probability distribution to offer diverse perspectives, many-to-one matching, and then evaluate their similarity with category text embedding via Monte-Carlo estimation.

Furthermore, to learn distinctive embedding space, we propose a distribution contrastive learning scheme to capture the statistical similarity between distributions. We enhance intra-class compactness by learning the similarity of content (action or background)

within videos and maximize inter-class separability by leveraging action category information across videos. To enhance the intra-class compactness, we draw inspiration from snippet mining in prior work [57] to differentiate a similar snippet distribution among related content. For inter-class separability, we build a video-level probabilistic distribution based on Gaussian mixture model (GMM) and make the mixture distribution separable between different action classes. To the best of our knowledge, this is the first attempt to investigate multimodal probabilistic representations for weakly supervised temporal action localization.

Our main contributions to this work are summarized as follows:

(1) We introduce a novel framework that aligns VLP knowledge and action knowledge within a probabilistic space to make full consideration of temporal dynamics for fine-grained motion modeling.

(2) We introduce an intra- and inter-distribution contrastive strategy to construct distinctive probabilistic embedding space.

(3) We conduct extensive experiments and ablation studies to reveal the significance of probabilistic embedding and the proposed method, and our superior performance on two public benchmarks (THUMOS14 and ActivityNet v1.3).

## 2 RELATED WORK

### 2.1 Weakly Supervised Temporal Action Localization

Weakly supervised temporal action localization (WTAL) is proposed to alleviate the laborious annotation procedure for Temporal Action Localization, training with only video-level labels. In the early stages of research, Multiple Instance Learning (MIL)-based approaches [10, 14, 21, 22, 33, 37, 46] were proposed, which pretended a video as a bag of multiple action and background instances. Zhang *et al.* [57] introduced a snippet contrast loss, refining the representation of ambiguous instances in the feature space through snippet mining and contrastive learning. Afterward, several approaches [12, 25, 42, 65] generate snippet-level pseudo labels to explicitly guide the model as a localization-by-localization framework. However, pseudo labels generated based on video-level supervision were inaccurate and noisy, making it challenging to achieve the desired performance.

Recently, approaches utilizing the semantic information of action category names have emerged to address the fundamental absence of temporal annotation in WTAL [17, 24]. Li *et al.* [24] designed a novel framework with a discriminative objective to enlarge inter-class differences and a generative objective to enhance intra-class integrity via text information. Chen *et al.* [17] proposed a novel distillation and collaboration framework with complementary Classification Based Pre-training (CBP) and Vision-Language Pre-training (VLP) branches. While these works distinguish themselves with promising performances without additional annotation costs, there is yet potential for further development. In our framework, we integrate VLP knowledge and human action knowledge within the probabilistic space previously unexplored in existing literature, enabling a better understanding of human action.

## 2.2 Vision Language Pre-training

Vision language pre-training (VLP) learns a joint representation through large-scale image-text pair datasets with consistent contextual information. A representative work is CLIP [39], mapping image-text pairs with consistent contextual information into the visual and textual encoders separately and facilitating the learning of a joint embedding space through aligned representations. CLIP has shown great success in many image understanding tasks, including image classification [6, 34], semantic segmentation [20, 40], image generation [7, 43], and visual question answering [35]. Building upon the success of CLIP in the image domain, some research efforts [29–31, 54] aiming to leverage the vision-language representation of CLIP in the video domain have emerged. Our work is also a contribution to the research aimed at extending VLP knowledge into the realm of untrimmed video and human action understanding.

## 2.3 Probabilistic Representation

The main idea of probabilistic embedding is to map inputs to probability distributions in the embedding space. To achieve this objective, the desired distributions are estimated by a deep neural network and optimized to maximize their likelihood. PCME [5] represents one-to-many relationships in the joint embedding space with uncertainty estimation and introduces a soft cross-modal contrastive loss. Park *et al.* [36] proposed self-supervised video representation learning that bridges contrastive learning with probabilistic embedding with Gaussian mixture model. ProbVLM [50] utilizes a probabilistic adapter that, without the requirement of extensive datasets or intensive computing, estimates probability distributions for the embeddings of a vision-language pre-trained model through inter- and intra-modal alignment in a post-hoc manner. The objective of this study is to transfer the knowledge of a pre-trained vision-language model into the probabilistic embedding space, with an explicit objective of strengthening human action understanding.

## 3 METHOD

In this section, we provide a detailed explanation of the proposed approach. We first describe our baseline approach in Sec 3.1. We then introduce the probabilistic embedding space with VLP knowledge for WTAL problem in Sec 3.2 and discuss the proposed intra- and inter-distribution contrastive learning in Sec 3.3. An illustration of the overall framework is presented in Figure 2.

### 3.1 Base Approach

*3.1.1 Problem Definition.* In the WTAL problem setting, a set of untrimmed videos $\{V_i\}_{i=1}^N$ and their corresponding video-level category labels $\{\mathbf{y}_i\}_{i=1}^N$ are given. Specifically, the video-level category label is a multi-hot vector $\mathbf{y}_i \in \{0,1\}^{C+1}$, where $C$ is the number of action classes. Since all untrimmed videos have various background regions, we add an auxiliary class to model the background. Due to the computational memory limit, we split each video into multi-frame, non-overlapping snippets and sample a fixed number $T$ of snippets to address the large variations in the video length. WTAL aims to predict a sequence of actions $\{c_i, s_i, e_i, p_i\}_{i=1}^M$ for an input video $V_n$, where $M$ is the number of proposals, $c_i$ is the action

category, $s_i$ and $e_i$ are the start and end time of each proposal, and $p_i$ is confidence score.

*3.1.2 Base method.* Following prior works [10, 21, 57], it is common practice to use a pre-trained extractor for snippet representation. We extract the RGB features $\mathbf{X}^R = \{\mathbf{x}_t^r\}_{t=1}^T$ and Optical Flow features $\mathbf{X}^O = \{\mathbf{x}_t^o\}_{t=1}^T$, which allows us to model temporal dynamics efficiently when embedding $D$-dimensional features $\mathbf{x}_t^r \in \mathbb{R}^D$ and $\mathbf{x}_t^o \in \mathbb{R}^D$ for each snippet. Afterwards, we concatenate features from each modality $[\mathbf{X}^R; \mathbf{X}^O] \in \mathbb{R}^{T \times 2D}$ and feed into base WTAL head $f_{base}$ to generate base feature $\mathbf{X}^B \in \mathbb{R}^{T \times 2D}$, written as:

$$\mathbf{X}^B = f_{base}([\mathbf{X}^R; \mathbf{X}^O]; \phi_{base}) \in \mathbb{R}^{T \times 2D}, \quad (1)$$

where $f_{base}$ is mainly implemented with a series of temporal convolution with ReLU activation. In addition, an attention weight $\mathbf{a} \in \mathbb{R}^{T \times 1}$ is generated to differentiate between the foreground and the background region:

$$\mathbf{a} = \frac{\mathcal{A}(\mathbf{X}^R, \mathbf{X}^O) + \mathcal{A}(\mathbf{X}^O, \mathbf{X}^R)}{2} \in \mathbb{R}^{T \times 1}, \quad (2)$$

where $\mathcal{A}(\cdot)$ is an attention branch consisting of several temporal convolutional layers. Following the MIL framework, we feed the base feature $\mathbf{X}^B$ to classification head $f_{cls}$ to generate the base class activation sequence (CAS) is defined as:

$$\mathbf{S}^{base} = f_{cls}(\mathbf{X}^B; \phi_{cls}) \in \mathbb{R}^{T \times (C+1)}. \quad (3)$$

Then we aggregate snippet-level activation scores, to obtain video-level class prediction $\mathbf{p}^{base} = \mathcal{K}(\mathbf{S}^{base}) \in \mathbb{R}^{C+1}$, where $\mathcal{K}(\cdot)$ denotes the top-k average pooling along the temporal axis. After obtaining the video-level category prediction, we can build a loss function $\mathcal{L}_{base}$ with cross-entropy loss as follows:

$$\mathcal{L}_{base} = -\sum_{c=1}^{C+1} \mathbf{y}^{base} \log(\mathbf{p}_c^{base}), \quad (4)$$

where $\mathbf{y}^{base} = [y_1, \cdots, y_C, 1] \in \mathbb{R}^{C+1}$ is the video-level label with an auxiliary background class. Meanwhile, background suppressed CAS is acquired by multiplying the attention weight $\mathbf{S}_{supp} = \mathbf{a} \otimes \mathbf{S}^{base}$. We can also build a loss function $\mathcal{L}_{supp}$ with background suppressed video-level class score $\mathbf{p}^{supp} = \mathcal{K}(\mathbf{S}_{supp}) \in \mathbb{R}^{C+1}$ as follows:

$$\mathcal{L}_{supp} = -\sum_{c=1}^{C+1} \mathbf{y}^{supp} \log(\mathbf{p}_c^{supp}), \quad (5)$$

where $\mathbf{y}^{supp} = [y_1, \cdots, y_C, 0] \in \mathbb{R}^{C+1}$ is the video-level label without a background class. By optimizing $\mathcal{L}_{cls} = \mathcal{L}_{base} + \mathcal{L}_{supp}$, the model learns to distinguish snippets that contribute significantly to video-level action classification. Moreover, we also utilized $\mathcal{L}_{oppo}$, $\mathcal{L}_{norm}$, $\mathcal{L}_{guide}$. Since these losses have been proposed in previous works [14, 21, 22, 32, 37], we do not claim our contribution in this part. The overall objective of baseline approach $\mathcal{L}_{vid}$ is as follows:

$$\mathcal{L}_{vid} = \lambda_1 \mathcal{L}_{cls} + \lambda_2 \mathcal{L}_{oppo} + \lambda_3 \mathcal{L}_{norm} + \lambda_4 \mathcal{L}_{guide}. \quad (6)$$

## 3.2 Probabilistic Class Activation Sequence

In this section, we reformulate CAS as a probabilistic class activation sequence (P-CAS) designed to effectively leverage VLP knowledge within a probabilistic embedding space. To achieve this,

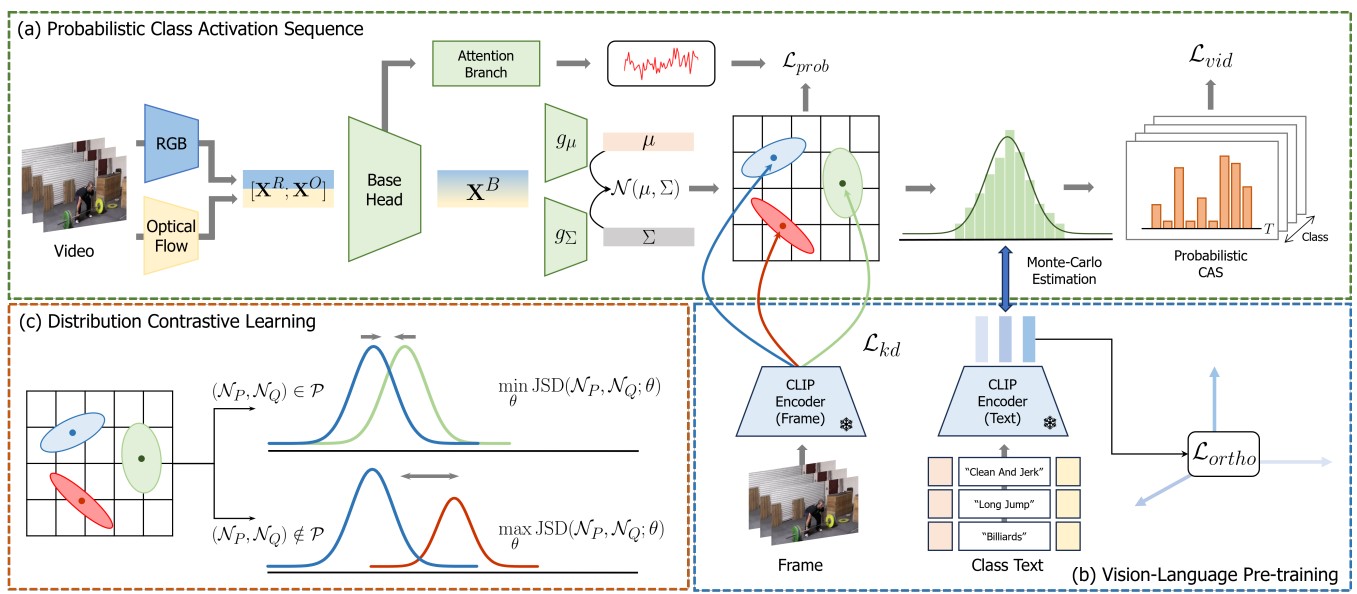

**Figure 2: Overview of the proposed PVLR. (a) Probabilistic Class Activation Sequence: For the probabilistic embedding, probabilistic adapters are augmented to facilitate the estimation of probabilistic distributions for individual snippets. (b) Vision-Language Pre-training: Leveraging VLP knowledge, we estimate probabilistic distributions and guide the model with semantic textual information corresponding to action categories. (c) Distribution Contrastive Learning: By training statistical similarities from probabilistic distribution, we aim to build distinctive embedding space.**

we model the probabilistic distribution $p_{\mathbf{z}|\mathbf{x}}(\mathbf{z}|\theta)$ and estimate the parameters $\theta$, optimizing neural networks via human action and VLP knowledge.

*3.2.1 Probabilistic Embedding.* Initially, we establish probabilistic embedding space by leveraging the human action pre-trained knowledge on Kinetics [2]. From the base feature $\mathbf{X}^B = \{\mathbf{x}_t\}_{t=1}^T \in \mathbb{R}^{T \times 2D}$, we formulate a snippet-level probability distribution $p(\mathbf{z}|\mathbf{x}_t)$ as a multivariate Gaussian distribution with a mean vector and a diagonal covariance matrix to model the probabilistic embedding space:

$$p(\mathbf{z}|\mathbf{x}_t) \approx \mathcal{N}(g_\mu(\mathbf{x}_t), \text{diag}(g_\Sigma(\mathbf{x}_t))), \qquad (7)$$

where $g_\mu$ is an embedding layer that estimates the mean vector $g_\mu(\mathbf{x}_t) \in \mathbb{R}^D$ and $g_\Sigma$ is an embedding layer that estimates covariance matrix $g_\Sigma(\mathbf{x}_t) \in \mathbb{R}^D$ of targeted Gaussian. With the estimated $p(\mathbf{z}|\mathbf{x}_t)$, we can sample $K$ random embeddings $\mathbf{z}^{(k)} \in \mathbb{R}^D$ that can represent the estimated distribution following [19]:

$$\mathbf{z}_t^{(k)} = g_\mu(\mathbf{x}_t) + \epsilon^{(k)} \cdot g_\Sigma(\mathbf{x}_t) \in \mathbb{R}^D, \qquad (8)$$

where $\epsilon^{(k)} \in \mathbb{R}^D$ are independently and identically sampled from a $D$-dimensional unit Gaussian. Our goal is to utilize $K$ embeddings sampled from the estimated probability distribution for each snippet to capture the human action from a more diverse range of perspectives. Also, for textual information, we transform action category names into pre-trained embeddings. For this, we freeze the CLIP text transformer $\Psi_C(\cdot)$ and extract the embeddings $\mathbf{X}^C = \{\mathbf{x}_c\}_{c=1}^C$:

$$\mathbf{x}_c = \Psi_C([\mathbf{L}_s; \Psi_{emb}(t_c); \mathbf{L}_e]) \in \mathbb{R}^D, \qquad (9)$$

where $\mathbf{L}_s, \mathbf{L}_e$ are learnable tokens, $t_c$ refers to action category, and $\Psi_{emb}$ is word embedding layer. Then, P-CAS can be defined by determining the action confidence score along the temporal axis with the estimated probability distribution and action category representation. Specifically, to measure the confidence score between the estimated distribution and category representation, we formulate P-CAS as $\mathbf{S}_{prob} \in \mathbb{R}^{T \times (C+1)}$ via Monte-Carlo estimation:

$$s_{prob}(t, c) \approx \frac{1}{K} \sum_{k=1}^K \text{sim}(\mathbf{x}_c, \mathbf{z}_t^{(k)})/\tau, \qquad (10)$$

where $\text{sim}(\cdot)$ means cosine similarity and $\tau$ is temperature parameter. Besides, to avoid ambiguity among action categories, we design an orthogonal loss $\mathcal{L}_{ortho}$ ensuring the uniqueness of each category representation as:

$$\mathcal{L}_{ortho} = \left\| \mathbf{X}^C (\mathbf{X}^C)^\top - \mathbf{I} \right\|_F^2, \qquad (11)$$

where $\mathbf{I}$ is the identity matrix and $\|\cdot\|_F^2$ is the Frobenius norm of a matrix.

*3.2.2 VLP Knowledge Distillation.* Nevertheless, during the estimation of the present probability distribution, only the textual information from VLP is utilized, overlooking the alignment between human action knowledge and the visual representation provided by VLP. Therefore, we aim to integrate VLP visual knowledge into the estimation process of the probability distribution.

Due to deterministic pre-training of CLIP, estimating the entire distribution can be challenging, but large-scale pre-trained representation can offer a generalized point approximation (*e.g.*, mean vector) for the desired distribution. To achieve this, our probabilistic

embedding utilizes the CLIP's deterministic representations as estimates for the mean, $g_\mu(\mathbf{x}_t)$ of the targeted distribution. To transfer VLP knowledge into probabilistic embedding space, we first sample a set of frames $\{f_t\}_{t=1}^T$ within a fixed temporal stride from the video. Next, we freeze the CLIP Image Encoder $\Psi_{\mathcal{I}}(\cdot)$ and extract the embeddings $\mathbf{X}^{\mathcal{I}} = \{\Psi_{\mathcal{I}}(f_t)\}_{t=1}^T \in \mathbb{R}^{T \times D}$. For a pair of snippet feature and CLIP image feature $(\mathbf{x}_t^b, \mathbf{x}_t^i)$, the distillation loss $\mathcal{L}_{kd}$ is defined as:

$$\mathcal{L}_{kd} = -\frac{1}{T} \sum_{t=1}^T \log\left(\frac{1}{2}\left(\frac{g_\mu(\mathbf{x}_t) \cdot \mathbf{x}_t^i}{\|g_\mu(\mathbf{x}_t)\| \|\mathbf{x}_t^i\|} + 1\right)\right). \quad (12)$$

We utilize the rescaled cosine similarity between the estimated mean $g_\mu(\mathbf{x}_t^b)$ and CLIP image representation $\mathbf{x}_t^i$ as the matching score. The objective of $\mathcal{L}_{kd}$ is to align the estimated mean $g_\mu(\mathbf{x}_t^b)$ with the generalized fixed point $\mathbf{x}_t^i$ of CLIP embedding for transferring pre-trained CLIP knowledge into the desired probability distribution.

### 3.3 Distribution Contrastive Learning

We defined a probabilistic embedding space by aligning human action knowledge and VLP knowledge. However the crucial factor of distributional similarity remained unexplored. The distributions corresponding to human actions should exhibit similarities amongst themselves while contrasting with background distributions. To address these objectives, we aim to enhance the completeness of the probabilistic embedding space through probabilistic representation learning based on statistical distances.

*3.3.1 Intra-Distribution Contrastive Learning.* We begin by considering contrastive learning between distributions within the video. In the video, the distributions of actions are expected to share distributional similarities internally while being separate from the background. To achieve these objectives, we adopt the snippet mining algorithm of zhang *et al.* [57], which uses attention weight $\mathbf{a} \in \mathbb{R}^{T \times 1}$ to differentiate action and background snippet within the video. Firstly, to mine the action and background snippet, we threshold the attention weight (1 or 0 indicates the action or background, respectively):

$$\mathbf{b}^{(t)} = \begin{cases} 1 & \text{if } \mathbf{a}^{(t)} > \theta_b, \\ 0 & \text{otherwise} \end{cases}, \quad (13)$$

where $\theta_b$ is the threshold value.

We utilize the same strategy performing, cascaded dilation or erosion operations to identify challenging samples (hard to differentiate) at the action/background boundaries.

$$\mathcal{R}_{inner} = (\mathbf{b}; m)^- - (\mathbf{b}; \mathcal{M})^- \quad (14)$$

$$\mathcal{R}_{outer} = (\mathbf{b}; \mathcal{M})^+ - (\mathbf{b}; m)^+, \quad (15)$$

where $(\cdot)^-$ and $(\cdot)^+$ represent the binary erosion and dilation operations with mask respectively. Following earlier work [57], we consider the inner regions as hard action snippet sets, and the outer regions are considered as hard background snippet sets. Here, we define hard actions as having a positive relation $\mathcal{P}_{act}$ with easy actions (top-k). Similarly, we define hard backgrounds as having a positive relation $\mathcal{P}_{bkg}$ with easy backgrounds (bottom-k). Note that we do not claim technical contribution over snippet mining. Instead,

our main contribution is to develop effective intra-distribution contrastive learning for probabilistic embedding space. After snippet mining, we utilize KL divergence as a statistical metric for measuring the similarity of snippet distributions. The KL divergence between multivariate Gaussian is defined as:

$$\text{KL}(\mathcal{N}_P \parallel \mathcal{N}_Q) = \frac{1}{2}(\text{tr}(\Sigma_Q^{-1}\Sigma_P) +$$

$$(\mu_Q - \mu_P)^{\mathrm{T}}\Sigma_Q^{-1}(\mu_Q - \mu_P) + \ln\left(\frac{\det \Sigma_Q}{\det \Sigma_P}\right) - D). \quad (16)$$

For intra-class compactness of embedding space, we propose an intra-contrastive loss $\mathcal{L}_{intra}$ to refine the snippet-level distribution similarity. Finally, the intra-contrastive loss $\mathcal{L}_{intra}$ is formulated as:

$$\mathcal{L}_{intra} = \begin{cases} -\log(1 - p(\mathcal{N})) & \text{if } (\mathcal{N}_P, \mathcal{N}_Q) \in \mathcal{P} \\ -\log(p(\mathcal{N})) & \text{otherwise} \end{cases}. \quad (17)$$

For the matching probability $p(\mathcal{N})$, we simply formulated as the Jensen-Shannon divergence $\text{JSD}(\mathcal{N}_P, \mathcal{N}_Q)$ to decrease the divergence of distributions corresponding for the positive pairs and maximize the divergence for the negative pairs.

*3.3.2 Inter-Distribution Contrastive Learning.* We further introduce inter-distribution contrastive learning utilizing action category labels to ensure inter-class separability. Here, we represent the whole video distribution $p(\mathbf{z}|V)$ as a Gaussian mixture model (GMM) to measure video-level similarity,

$$p(\mathbf{z}|V) \approx \sum_{t=1}^T \mathbf{a}_t \cdot \mathcal{N}(g_\mu(\mathbf{x}_t), \text{diag}(g_\Sigma(\mathbf{x}_t))). \quad (18)$$

For estimating $p(\mathbf{z}|V)$, we adopt the attention weight $\mathbf{a} \in \mathbb{R}^T$ as a mixing coefficient to appropriately mix distributions based on the actionness score. Given video-level category labels, we formulate a self-similarity map $\mathbf{H} \in \mathbb{R}^{N \times N}$ (1 for the same class, 0 for different) to characterize relationships between videos. Similar to intra-contrastive learning, we compute the matching probabilities between $N$ videos within a mini-batch across mixture models, then enhance inter-video representation by comparing them with the self-similarity map. Finally, the inter-contrastive loss $\mathcal{L}_{inter}$ is formulated as:

$$\mathcal{L}_{inter} = -\frac{1}{N^2} \sum_{i=1}^N \sum_{j=1}^N \mathcal{L}_{\text{BCE}}(\mathbf{H}(i, j), p(\mathcal{N})), \quad (19)$$

where $\mathcal{L}_{\text{BCE}}$ is a binary cross entroy loss.

Beyond aligning with VLP knowledge and the probabilistic embedding space, our proposed contrastive learning framework poses constraints on the suggested probabilistic representation to ensure compliance with both intra-compactness and inter-separability.

*3.3.3 Total Objectives.* Given all the previously mentioned objectives, the total objective $\mathcal{L}_{total}$ of the entire framework is determined as:

$$\mathcal{L}_{total} = \mathcal{L}_{vid} + \alpha\mathcal{L}_{kd} + \beta\mathcal{L}_{ortho} + \gamma\mathcal{L}_{prob}, \quad (20)$$

where $\alpha, \beta, \gamma$ are the hyper-parameters to balance these loss terms and $\mathcal{L}_{prob} = \mathcal{L}_{intra} + \mathcal{L}_{inter}$.

**Table 1: Comparison with previous state-of-the-art methods on THUMOS14. 0.1:0.7 and 0.1:0.5 represent the average mAP under IoU thresholds of 0.1:0.7 and 0.1:0.5.**

| Supervision | Method | Venue | mAP@IoU (%) | | | | | | | AVG | |
|---|---|---|---|---|---|---|---|---|---|---|---|
| | | | 0.1 | 0.2 | 0.3 | 0.4 | 0.5 | 0.6 | 0.7 | 0.1:0.7 | 0.1:0.5 |
| Fully supervised | TAL-Net [3] | CVPR 2018 | 59.8 | 57.1 | 53.2 | 48.5 | 42.8 | 33.8 | 20.8 | 45.1 | 52.3 |
| | P-GCN [56] | CVPR 2019 | 69.5 | 67.8 | 63.6 | 57.8 | 49.1 | - | - | - | 61.6 |
| | BUMR [62] | ECCV 2020 | - | - | 53.9 | 50.7 | 45.4 | 38.0 | 28.5 | - | - |
| Weakly supervised | CoLA [57] | CVPR 2021 | 66.2 | 59.5 | 51.5 | 41.9 | 32.2 | 22.0 | 13.1 | 40.9 | 50.3 |
| | $CO_2$-Net [10] | MM 2021 | 70.1 | 63.6 | 54.5 | 45.7 | 38.3 | 26.4 | 13.4 | 44.6 | 54.4 |
| | Xu *et al.* [53] | TPAMI 2023 | 73.1 | 66.9 | 58.3 | 48.8 | 36.5 | 24.4 | 13.4 | 45.9 | 56.7 |
| | Li *et al.* [24] | CVPR 2023 | 71.1 | 65.0 | 56.2 | 47.8 | 39.3 | 27.5 | 15.2 | 46.0 | 55.9 |
| | ASCN [63] | TMM 2023 | 71.4 | 65.6 | 57.0 | 48.2 | 39.8 | 26.8 | 14.4 | 46.2 | 56.4 |
| | Wang *et al.* [52] | CVPR 2023 | 73.0 | 68.2 | 60.0 | 47.9 | 37.1 | 24.4 | 12.7 | 46.2 | 57.2 |
| | SMEN [47] | TCSVT 2023 | 74.0 | 68.5 | 60.1 | 49.4 | 36.9 | 23.6 | 12.9 | 46.5 | 57.8 |
| | Li *et al.* [23] | TNNLS 2023 | 71.7 | 66.9 | 57.2 | 48.0 | 40.4 | 27.5 | 14.4 | 46.6 | 56.8 |
| | Zhang *et al.* [60] | TCSVT 2023 | 72.6 | 67.1 | 59.5 | 49.3 | 39.4 | 26.5 | 13.4 | 46.8 | 57.6 |
| | P-MIL [41] | CVPR 2023 | 71.8 | 67.5 | 58.9 | 49.0 | 40.0 | 27.1 | 15.1 | 47.0 | 57.4 |
| | LPR [11] | TCSVT 2023 | 71.9 | 66.7 | 57.4 | 48.4 | 40.3 | 28.5 | 15.8 | 47.0 | 56.9 |
| | AHLM [51] | ICCV 2023 | **75.1** | 68.9 | 60.2 | 48.9 | 38.3 | 26.8 | 14.7 | 47.2 | 58.3 |
| | DDG-Net [48] | ICCV 2023 | 72.5 | 67.7 | 58.2 | 49.0 | 41.4 | 27.6 | 14.8 | 47.3 | 57.8 |
| | STCL-Net [8] | TPAMI 2023 | 72.7 | 67.1 | 58.2 | 49.7 | 41.8 | 28.7 | 16.0 | 47.7 | 57.9 |
| | GauFuse [65] | CVPR 2023 | 74.0 | 69.4 | 60.7 | 51.8 | 42.7 | 26.2 | 13.1 | 48.3 | 59.7 |
| | Ju *et al.* [17] | CVPR 2023 | 73.5 | 68.8 | **61.5** | **53.8** | 42.0 | 29.4 | 16.8 | 49.4 | 60.0 |
| | Zhang *et al.* [61] | TMM 2024 | 71.1 | 65.0 | 56.4 | 46.6 | 38.0 | 26.1 | 13.0 | 45.2 | 55.4 |
| | SPCC-Net [44] | TMM 2024 | 72.6 | 67.3 | 59.4 | 48.7 | 38.3 | 25.6 | 13.4 | 46.5 | 57.3 |
| | Yun *et al.* [55] | AAAI 2024 | 72.4 | 66.9 | 58.4 | 49.7 | 41.8 | 25.5 | 12.8 | 46.8 | 57.8 |
| | SRHN [64] | TCSVT 2024 | 73.1 | 67.1 | 58.3 | 49.6 | 40.8 | 28.2 | 14.1 | 47.3 | 57.8 |
| | **PVLR** (Ours) | - | 74.9 | **69.9** | 61.4 | 53.1 | **45.1** | **30.5** | **17.1** | **50.3** | **60.9** |

# 4 EXPERIMENTS

## 4.1 Experimental Settings

We conduct experiments on two popular WTAL benchmarks: THU-MOS14 [13] and ActivityNet v1.3 [1]. THUMOS14 is a widely used benchmark for the WTAL problem. It contains 200 validation videos and 213 test videos for 20 sports categories. Following previous works [10, 42, 57], we use 200 validation videos to train our framework and use 213 test videos for evaluation. In WTAL, THUMOS14 is the most challenging dataset because of the motion blur, significant intra-class varieties, and extremely short action instances. ActivityNet v1.3 has 10,024 training videos, 4,926 validation videos, and 5,044 testing videos from 200 action categories. Since annotations for the testing set are not released, we train on the training set and test on the validation set. Challenges in ActivityNet usually lie in numerous action categories. Following the standard evaluation metrics, we evaluate our method with mean Average Precision (mAP) under different Intersection over Union (IoU) thresholds on the temporal axis. We adopt the same evaluation code from previous works [10, 14] for fair comparisons. Furthermore, in the inference phase, we computed the action confidence score using the estimated mean without utilizing the reparameterization strategy.

## 4.2 Implementation Details

To conduct experiments on the THUMOS14 dataset and ActivityNet v1.3 dataset, we first divide each video into non-overlapping segments consisting of 16 frames. Subsequently, we extract the 1024-dimensional RGB and optical flow features from the I3D network [2] pre-trained on the Kinetics400 dataset. We utilized the TV-L1 algorithm to extract the flow features. The fixed number of segments $T$ is set to 320 and 60 for THUMOS14 and ActivityNet v1.3, respectively. For the base WTAL model, it can be any existing WTAL method and we use the $CO_2$-Net [10] as the base head for its simple framework. We adopt ResNet-50 as a backbone network for the CLIP image encoder $\Psi_{\mathcal{I}}$. It is worth noting that the I3D network and the CLIP encoders are not fine-tuned during training. For CLIP image feature, we divide the video as described above and the middle frame of each snippet is fed into $\Psi_{\mathcal{I}}$. For the probabilistic adapter, $g_\mu$ indicates a single linear layer, while $g_\Sigma$ is a separate network with a linear layer followed by the ReLU function, to ensure the $\Sigma$ remains positive definite. Similar to previous work [16], we prepend and append 4 prompt vectors to word embedding $\Psi_{emb}(t_c)$, which is initialized with $\mathcal{N}(0, 0.01)$. In P-CAS, we used the learnable vector to model the background class, which is hard to characterize. Our experiments are conducted on an NVIDIA Tesla V100 GPU.

Table 2: Results on ActivityNet v1.3. 0.5:0.95 indicates the average mAP at IoU thresholds of 0.5:0.95.

| Method | Venue | mAP@IoU (%) | | | AVG |
|---|---|---|---|---|---|
| | | 0.5 | 0.75 | 0.95 | 0.5:0.95 |
| DCC [25] | CVPR 2022 | 38.8 | 24.2 | 5.7 | 24.3 |
| RSKP [12] | CVPR 2022 | 40.6 | 24.6 | 5.9 | 25.0 |
| ASM-Loc [9] | CVPR 2022 | 41.0 | 24.9 | 6.2 | 25.1 |
| STCL-Net [8] | TPAMI 2023 | 40.6 | 24.0 | 6.0 | 24.7 |
| Zhang *et al.* [60] | TCSVT 2023 | 41.6 | 25.1 | 6.5 | 25.3 |
| LPR [11] | TCSVT 2023 | 41.4 | 25.3 | 6.2 | 25.4 |
| P-MIL [41] | CVPR 2023 | 41.8 | 25.4 | 5.2 | 25.5 |
| AHLM [51] | ICCV 2023 | 42.3 | 24.8 | **6.9** | 25.9 |
| Li *et al.* [24] | CVPR 2023 | 41.8 | 26.0 | 6.0 | 26.0 |
| Wang *et al.* [52] | CVPR 2023 | 41.8 | 25.7 | 6.5 | 26.3 |
| Li *et al.* [23] | TNNLS 2023 | 42.3 | 26.4 | 6.1 | 26.4 |
| CASE [29] | ICCV 2023 | 43.2 | 26.2 | 6.7 | 26.8 |
| Yun *et al.* [55] | AAAI 2024 | 39.4 | 25.8 | 6.4 | 25.8 |
| SRHN [64] | TCSVT 2024 | 41.7 | 26.1 | 6.1 | 26.2 |
| Liu *et al.* [28] | ICASSP 2024 | 42.8 | 26.8 | 6.0 | 26.4 |
| **PVLR** (Ours) | - | **43.6** | **27.4** | 6.5 | **27.4** |

## 4.3 Comparison With State-Of-The-Art Methods

In this section, we compare our proposed PVLR with previous state-of-the-art methods. For the THUMOS14 [13], it is evident that the proposed PVLR outperforms the performance of all previous state-of-the-art methods as shown in Table 1. Especially in the WTAL scenarios, where performance under high IoU(0.5-0.7) is of particular importance, our proposed method distinguishes itself by surpassing the performance of all existing methodologies. In direct comparison to prior studies [17, 24] that also incorporate textual information, our approach exhibits superior performance, with a margin ranging from 0.9% to 4.3% when it comes to the important criterion of average mAP (0.1:0.7). Additionally, our approach was shown to either outperform or reach similar performance levels as recent fully supervised methods. In Table 2, results for the larger dataset ActivityNet v1.3 are presented. Similarly, our proposed PVLR demonstrates superior performance compared to existing weakly supervised state-of-the-art methods.

## 4.4 Ablation Study

To demonstrate the effectiveness of our model components, we analyze the impact of each component in this section with THU-MOS14 [13]. Table 3 presents the effect of the proposed components in comparison to the baseline approach [10]. The VLP knowledge distillation module serves as a pivotal step within our framework, marking the inception of our approach. By conducting feature alignment in a probabilistic space, PVLR introduces a fundamental basis that was previously overlooked in earlier literature. By integrating VLP knowledge, we realized a performance boost of 2.4% through the implementation of a probabilistic class activation sequence (P-CAS). Also, refinement of our probabilistic embedding space, which benefited from the introduction of distribution contrastive learning, led to a 2.0% improvement in efficacy. Finally, an evident gain of

Table 3: Component-wise ablation study on THUMOS14.

| Method | mAP@IoU | | | AVG |
|---|---|---|---|---|
| | 0.3 | 0.5 | 0.7 | 0.3:0.7 |
| Baseline [10] | 54.5 | 38.3 | 13.4 | 35.7 |
| +Distillation from VLP knowledge | 58.1 | 40.3 | 15.3 | 38.1 |
| +Intra-contrastive | 59.4 | 42.3 | 15.9 | 39.5 |
| +Inter-contrastive | 59.6 | 43.7 | 16.9 | 40.1 |
| +Orthogonalization of text prompts | **61.4** | **45.1** | **17.1** | **41.4** |

Table 4: Further analysis for probabilistic representation.

| Metric | mAP@IoU | | | AVG |
|---|---|---|---|---|
| | 0.3 | 0.5 | 0.7 | 0.3:0.7 |
| Deterministic CAS | 57.5 | 41.0 | 15.8 | 38.3 |
| Mahalanonis Distance | 60.1 | 44.0 | 17.3 | 40.7 |
| Bhattacharyya Distance | 60.8 | 44.0 | 17.3 | 40.9 |
| Kullback–Leibler Divergence | **61.4** | **45.1** | **17.1** | **41.4** |

1.3% can be obtained by introducing the orthogonalization of text embedding which enhances the discriminative capacity between text category embeddings. We have ultimately demonstrated the effectiveness of our proposed module by achieving a performance improvement of 5.7%, a level that has been difficult to find in previous research.

## 4.5 Discussion

To provide deeper insights into the design aspect of our proposed framework, we conducted several experiments in this section.

*4.5.1 Probabilistic Representation.* As the initiative procedure in our framework, the probabilistic representation is of great importance. However, to compare its significance, we developed a simple baseline using deterministic representation. For the deterministic baseline, we conducted experiments utilizing one-to-one matching between human action knowledge and text embedding without probabilistic adapter. In Table 4, the first row "Deterministic CAS" indicates the deterministic baseline. Quantitatively comparing, there is a performance decrease of about 3.1% compared to the proposed probabilistic approach in Table 4. Also, we compare qualitative visualization results of selected videos from the THU-MOS14 dataset. Figure 3 illustrates that the deterministic approach frequently produces predictions beyond the ground truth boundaries, struggling to capture subtle variations in human action. In contrast, the probabilistic method effectively models temporal dynamics, focusing its predictions on the segments where real actions unfold. In Figure 3(a), the probabilistic approach appears to struggle with completely filling the GT segment, yet this specific area, characterized by an absence of motion change, is designated for future exploration. From Table 4, besides the KL divergence, other statistical metrics are also suitable for our contrastive learning. Table 4 reveals that metrics capable of assessing inter-distributional similarity exhibit relatively consistent performance with minimal

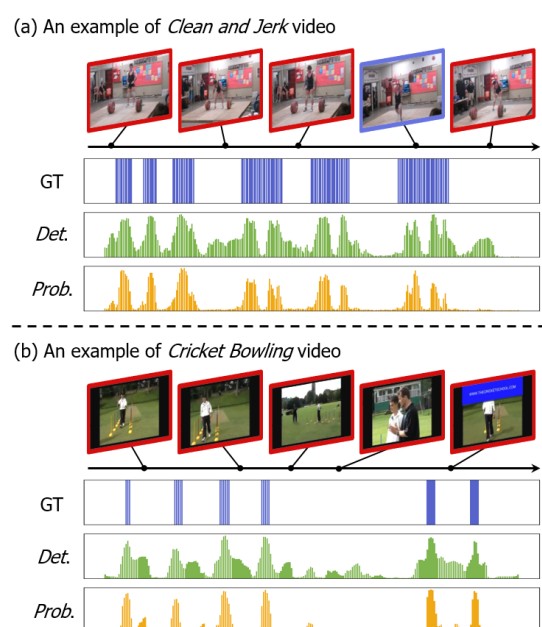

Table 5: Number of $K$ ablation study on THUMOS14.

| # of samples | mAP@IoU | | | AVG |
| | 0.3 | 0.5 | 0.7 | 0.3:0.7 |
| --- | --- | --- | --- | --- |
| Baseline [10] | 54.5 | 38.3 | 13.4 | 35.7 |
| $K = 0$ | 57.5 | 41.0 | 15.8 | 38.3 |
| $K = 5$ | 59.8 | 41.4 | 15.5 | 38.9 |
| $K = 10$ | 60.1 | 43.8 | 16.9 | 40.5 |
| $K = 20$ | **61.4** | **45.1** | **17.1** | **41.4** |

Table 6: Framework generalization results on THUMOS14.

| Method | mAP@IoU | | | AVG |
| | 0.3 | 0.5 | 0.7 | 0.3:0.7 |
| --- | --- | --- | --- | --- |
| BaS-Net [21] | 44.6 | 26.6 | 10.0 | 27.0 |
| BaS-Net+Ours | 50.1 | 29.2 | 10.7 | **30.2** |
| CoLA [57] | 51.8 | 34.0 | 12.5 | 32.9 |
| CoLA+Ours | 56.2 | 35.5 | 13.3 | **35.1** |

Figure 3: Qualitative Results on THUMOS14. We compared the class activation sequence (CAS) of deterministic and probabilistic approaches. In this case, the red box is for the background, and the blue box is for the action.

variation. By not relying on a specific distance metric, it can be considered that a well-generalized probability distribution has been estimated, leading to the successful modeling of a probabilistic embedding space. Finally, the marginal superiority of KL divergence led to its utilization for the proposed distribution contrastive learning.

*4.5.2 Number of K samples.* During the generation of P-CAS, we reparameterized $K$ samples of the estimated snippet distribution to compute the similarity between the estimated distribution and the action category text embedding. To analyze the impact of the number of samples, we compared the performances under different values of $K$, as shown in Table 5. As observed, a small value for $K$ leads to suboptimal performance, resulting in a lack of representation of the estimated distribution. Here, we denoted the previously described deterministic baseline as $K = 0$. Considering that larger values of $K$ capture the entire distribution of the snippet through Monte-Carlo estimation, performance improves with an increase in $K$. Nevertheless, an increased value for $K$ results in higher computational demands. Calculating the confidence score for composing a P-CAS requires computations on the order of $O(K)$ for each snippet and action category. Considering the computational overhead, we decided on $K = 20$.

## 4.6 Generalization Study

In Table 6, we demonstrate the generality of our contributions by integrating them into previous works in a plug-and-play manner.

To achieve this, we conducted comparative experiments by substituting the base WTAL head with previous works [21, 57]. The additional training modules exclusively considered the proposed probabilistic adapter for probabilistic embedding. Furthermore, we reformulated the classification objective using probabilistic class activation sequences (P-CAS). The results show that our framework enhances their performance by an average mAP increase ranging from 2% to 3%, indicating robust generalization across various methods and model architecture designs.

## 5 CONCLUSION AND FUTURE WORKS

In this work, we present a novel framework that leverages large-scale vision-language pre-training (VLP) to address weakly supervised temporal action localization (WTAL). Our motivation stemmed from the observation that the deterministic representation with VLP may not be optimal for WTAL. Furthermore, we noticed that previous studies did not consider the alignment between human action and VLP knowledge. To address these concerns, we introduce a probabilistic embedding framework aligned with human action and VLP knowledge, enhanced by distribution contrastive learning. Our method significantly outperforms previous approaches on two prominent datasets, revealing the efficacy of probabilistic embedding within the VLP representation. However, the exploration of probabilistic embedding for text data solely represented by action category names remains unexplored. For future work, we will explore leveraging the recently acclaimed large-language model (LLM) to generate attributes for each action category and subsequently integrate them with probabilistic embeddings. We believe that the probabilistic embedding with a vision-language pretrained model will be a promising direction for various weakly supervised and unsupervised learning tasks.

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
