# OpenReview forum: "Probabilistic Vision-Language Representation for Weakly Supervised Temporal Action Localization"
_acmmm.org/ACMMM/2024/Conference — MM2024 Poster_

### Official Review · Reviewer_3e2a · 2024-05-14

**Rating:** 4
**Confidence:** 4

**Summary:**

This work propose a novel framework that aligns human action knowledge and VLP knowledge in the probabilistic embedding space. Meawhile, it leverage intra- and inter-distribution contrastive learning to enhance the probabilistic embedding space based on statistical similarities. Extensive experiments prove that it significantly outperforms other works.

**Strengths:**

1. This work aligns VLP knowledge and action knowledge within a probabilistic space, which is technically-sound and novel.

2. This work  introduce an intra- and inter-distribution contrastive strategy to construct distinctive probabilistic embedding space.

3. This work  performs extensive experiments on two public datasets, demonstrating the outstanding performance of the model on various evaluation metrics, and contrasting and analyzing it with existing methods.

**Limitations:**

1.The Section 3.2 is include many loss function. The description of their interaction is not very obvious, it looks somewhat cumbersome and intricate, in the revision, the authors need to polish this part to make it clearer.

2. There are severa hyper-parameters like $\lambda$ in Eq.6, $\alpha$, $\beta$, $\gamma$ in Eq.20. What are the effects of using different values? Ablations on these hyper-parameters are missing.

3.The paper does not provide an analysis of the complexity and running time of the model, nor does it discuss the efficiency and scalability of the model, which makes the application performance of the model on other datasets unclear.

4. Compared with other results on THUMOS (table1), the performance is not well . What is the possible reason for this?

**Suitability:**

3

---

### Official Review · Reviewer_WFTv · 2024-05-19

**Rating:** 5
**Confidence:** 4

**Summary:**

This paper introduces a novel framework for Weakly Supervised Temporal Action Localization (WTAL), which integrates Vision-Language Pre-training (VLP) knowledge and human action knowledge within a probabilistic embedding space to better capture temporal dynamics and fine-grained human motion. Extensive experiments and ablation studies demonstrate the effectiveness of the proposed probabilistic embedding and the overall method, showing significant improvements over state-of-the-art methods.

**Strengths:**

1. The idea of constructing a probabilistic space using VLP knowledge and leveraging uncertainty information to achieve many-to-one alignment between textual information and fine-grained actions is novel.
2. Achieved new state-of-the-art (SOTA) results on multiple public datasets.

**Limitations:**

1. In addition to the I3D features, this method requires extracting CLIP features for each snippet during runtime, which significantly increases computational complexity, memory usage, and runtime.
2. The necessary of VLP konwledge Distillation: The visual representations of the VLP model already reflect action semantics to some extent. In the mean estimation of snippet-level distributions, why not directly use the mean of CLIP's visual representations, but instead distill CLIP knowledge into the mean estimator defined by the authors? What would be the impact of directly using CLIP visual feature representations as the mean?
3. The necessary of L𝑜𝑟𝑡ℎo: The purpose of leveraging textual information is to utilize additional textual data. When all category label features are orthogonal, it seems to be no different from using one-hot representations, as one-hot labels are a case where all category label representations are orthogonal. Could you conduct a comparative experiment to see what impact directly using one-hot representations instead of  X^C?

**Suitability:**

3

---

### Official Review · Reviewer_kM3X · 2024-05-24

**Rating:** 3
**Confidence:** 4

**Summary:**

The paper proposes a VLM-enhanced probabilistic representation learning method for Weakly Supervised Temporal Action Localization (WSTAL).
The text-encoder of CLIP is used to encoder the semantics of categories labels and an orthogonal loss is proposed to enhance the uniqueness representations for each category.
The visual-encoder of CLIP is used to encode visual input to ensure a good mean representation of the learned probabilistic embedding.
Distribution Contrastive Learning is proposed to ensure a well learned probabilistic embedding.

**Strengths:**

1. Probabilistic representations and corresponding Distribution Contrastive Learning is a novel idea to me for the Weakly Supervised Temporal Action Localization (WSTAL).
2. Experiments in commonly used datasets demonstrated the effectiveness of the proposed method, achieving the best performance.

**Limitations:**

1. The probabilistic representations are implemented by learning a mean and variances, which is not an inspired technology. And there are methods have exploration the uncertainty of representations, such as [1], which is also a probabilistic representation and using a distribution-based distance metric.
2. The method present is hard to read, and there are some missing explanations for symbol in the equation, such as $m, \mathcal{M}$ in Eq.(14) and Eq.(15), $\mathcal{P}, \mathcal{N}_p, \mathcal{N}_Q$ in Eq.(17).
3. The results of the paper achieve best compared with existing methods, but the paper additionally used a pre-trained CLIP, which is not fair for the existing methods that only use the I3D features. I am wondering the results of adding the proposed orthogonal loss of category representations and distillation loss of visual representations to existing method.
4. The results of table 6 can be integrated into table 3 that changing the backbone for ablation study to make the proposed method more convincing.

[1] Chang J, Lan Z, Cheng C, et al. Data uncertainty learning in face recognition[C]//Proceedings of the IEEE/CVF conference on computer vision and pattern recognition. 2020: 5710-5719.

**Suitability:**

3

---

### Official Review · Reviewer_s746 · 2024-05-24

**Rating:** 3
**Confidence:** 3

**Summary:**

This paper tackles the task of weakly supervised temporal action localization. The authors propose a probabilistic modeling framework that utilizes CLIP. Experiments conducted on the Thumos and ActivityNet datasets demonstrate that their methods achieve superior results compared to existing approaches.

**Strengths:**

1. The idea of replacing deterministic modeling with probabilistic approaches is interesting.

2. Experiments on the Thumos and ActivityNet datasets show improvement with the proposed methods.

**Limitations:**

**Major concerns**:

1. The final training objective actually contains an excessive number of loss terms (more than 9 items), including
$$
L_{kd}, L_{norm}, L_{ortho}, L_{oppo}, L_{guide}, L_{intra}, L_{inter}, L_{base}, L_{supp}
$$
and so forth. This multitude of terms complicates implementation, particularly in determining hyperparameters such as the loss weights for each term.  Are all nine of these items truly valuable? The author should consider eliminating redundant terms and simplifying the training objectives to enhance clarity.

2. In Table 3, the author reference [10] as their baseline. But a disparity emerges between [10] and the loss function delineated in Equation 6. Notably, the main contribution of [10] is to introduce a consensus loss, which is absent from this paper's loss function. Why do the metric values in Table 3 remain unchanged compared to those in reference [10]?

**Other concerns**:
1. Section 3.2 illustrates how to model $p(z \mid \theta)$. However, the variable $z$ is introduced here without any prior explanation. What does $z$ denotes here?

2. The paper uses the term "vision-language pretraining" (VLP) in a misleading way, indicating that VLP is part of their proposed methods. However, there is actually no VLP in their proposed method.  For instance, the following sentences inaccurately suggest that vision-language pretraining is part of their methods:
i) In the "Conclusion" section, the author claims they "leverage large-scale vision-language pre-training to address the WAL task.”
ii) In Figure 2 of the "Overview of Proposed Methods," the author labels part b of their method as "vision-language pretraining."
The author's intended meaning should be that they utilize "pretrained vision-language models" rather than conducting "vision-language pretraining" themselves.

3. The implementation details for hyperparameter configuration are lacking, specifically for parameters such as $\alpha$, $\beta$, and $\gamma$ in Eq. 20, and $\lambda_1$, $\lambda_2$, $\lambda_3$, $\lambda_4$ in Eq. 6.
Additionally, details for other hyperparameters like batch size, and learning rate are missing.

**Note**: I would raise my rating If my major concerns are addressed.

**Suitability:**

3

---

### Meta-Review · Area_Chair_KodS · 2024-07-04

**Recommendation:** Accept (Poster)
**Confidence:** 4

**Metareview:**

This paper proposes a probabilistic vision-language representation learning method for Weakly Supervised Temporal Action Localization, which uses the CLIP and distribution contrastive learning to achieve good alignments between human action knowledge and VLP knowledge in the probabilistic embedding space. In the rebuttal, the questions and concerns of reviewers have been addressed and most reviewers reach a consensus on the acceptance of this submission.

Quality: The idea of probabilistic representation learning on Temporal Action Localization is interesting and effective in weakly supervised settings.

Clarity: The paper is well-written, but the presentation about method details and illustration should be improved.

Originality: The novelty of this work is sufficient for acceptance.

Significance： The proposed framework has wide applications based on probabilistic vision-language representation learning and has the potential usage on exiting VLM downstream tasks.